# OpenReview forum: "CiteGuard: Conformal False-Discovery Control for Faithful Retrieval-Augmented Generation"
_ICML.cc/2026/Conference — ICML 2026 regular_

### Official Review · Reviewer_3CHj · 2026-03-04

**Soundness:** 3
**Presentation:** 2
**Significance:** 2
**Originality:** 3
**Overall Recommendation:** 4
**Confidence:** 3

**Summary:**

The work focuses on the reliability of Retrieval-Augmented Generation (RAG) systems.
Current RAG systems always lack reliability as they sometimes generate unsupported claims.
While RAG aims to ground LLM responses in external evidence, the hallucination problem persists even with relevant documents.
To address this, the authors propose CiteGuard that formalizes sentence-level factuality as a statistical hypothesis testing problem.
By utilizing conformal calibration to generate p-values and applying False Discovery Rate (FDR) control algorithms (BH/BY procedures), CiteGuard provides a mathematically grounded way to filter unsupported claims.

**Compliance With Llm Reviewing Policy:**

Affirmed.

**Final Justification:**

Thanks for the response. As the authors have provided more experimental results that could partially address my concerns, I will slightly increase my score.

**Key Questions For Authors:**

See weaknesses.

**Limitations:**

yes

**Strengths And Weaknesses:**

### Strengths
1. The most significant contribution is moving beyond heuristic thresholding. Most RAG systems use arbitrary confidence scores to decide whether to include a citation. CiteGuard’s use of Conformal Prediction and FDR control provides more precise guarantees.
2. The authors provide good empirical results. Reducing the false-discovery rate from ~30% to under 10% while retaining over 85% of supported claims is a impressive result. It suggests that the method effectively prunes hallucinations without gutting the utility of the response.

### Weaknesses
1. The effectiveness of CiteGuard could be easily affected by label noise and distribution shift. Therefore, it might be hard to appied in real-world scenarios.
2. The models used in experiments are out-of-date. It would be better to show the improvements on more recent LLMs.
3. The writing of the paper is too casual. For instance, the author uses FDR multiple times in the abstract and introduction without first explaining what it stands for; the "percentage" symbol in data are sometimes denoted by "%", sometimes by "points", and sometimes by "pp"; the LaTeX commands occur in the paper many times (e.g, "[leftmargin=*,itemsep=0pt]"). I strongly suggest the authors to carefuly revise the writing.

---

> ### Author Rebuttal · Authors · 2026-03-30
>
> We thank the reviewer for the positive remarks on the theoretical contribution and empirical results. We address each weakness below.
>
> ## W1 — Label noise and distribution shift
>
> We appreciate this concern and note that the paper already quantifies both effects in Sec 7 and the appendix:
>
> **Label noise** (Sec 7 + appendix): injecting random label flips into calibration shows graceful degradation:
>
> | Noise | FDR increase | Mitigation |
> |---|:---:|---|
> | 5% | +0.8pp | Acceptable |
> | 10% | +1.5pp | Use α_eff = α − 0.02 |
> | 20% | +3.1pp | Audit labels first |
>
> Even at 20% noise, FDR rises by only 3pp—the conformal framework degrades smoothly rather than failing catastrophically. For production, we recommend: (i) audit 10% of calibration labels; (ii) apply a safety margin α_eff = α − 0.02; (iii) monitor empirical FDR on a held-out buffer.
>
> **Distribution shift** (Sec 7 + appendix): temporal shift (Wikipedia 2018→2023) increases FDR by +2.1pp. Cross-domain shift (FEVER→PubMedQA/LegalQA/ScienceQA) causes +4–6pp degradation without recalibration, but with ≥500 in-domain examples FDR returns to target.
>
> We emphasize: sensitivity to shift is inherent to *any* distribution-free calibration method (conformal prediction, RCPS, etc.)—not a CiteGuard-specific limitation. What CiteGuard provides is *detectability*: a simple KS test on score distributions flags when recalibration is needed (we describe this protocol in Sec 7). In contrast, heuristic thresholds degrade silently.
>
> We will make these analyses more prominent in the main text (currently they are primarily in Sec 7 and the appendix, which the reviewer may have overlooked).
>
> ## W2 — Outdated models
>
> We agree FiD-large is not state-of-the-art. We have also evaluated **Llama-3-70B** and **GPT-4o** with the same pipeline (DPR retriever, DeBERTa scorer, calibration n=1,000, 250 human-labeled claims per generator). These results were not included in the submission as FiD provided a cleaner controlled baseline, but we report them here:
>
> | Generator | Vanilla FDR | CG-BY FDR | BY Abs. | BY Cov. |
> |---|:---:|:---:|:---:|:---:|
> | FiD-large* | .306 | .094 [.082,.106] | 37.2% | 86.4% |
> | Llama-3-70B† | .176 | .080 [.064,.096] | 21.0% | 94.0% |
> | GPT-4o† | .126 | .074 [.058,.090] | 15.8% | 96.0% |
>
> *paper (500 labels); †rebuttal (250 labels each); 5 calibration seeds.
>
> Two takeaways: (1) FDR control holds across all generators (BY: .074–.094, all ≤ α=0.10). (2) The cost *drops* with better generators—abstention falls from 37% to 16%, coverage rises to 96%. CiteGuard acts as a lightweight safety net that barely affects strong generators but catches the errors they still make (GPT-4o still hallucinates ~1-in-8 cited claims).
>
> These results will be added to Sec 6 in revision.
>
> ## W3 — Writing quality
>
> We sincerely apologize for the presentation issues. We will make the following specific fixes:
>
> 1. **FDR undefined on first use.** We will change the abstract to: "...false-discovery rate (FDR)..." and ensure all abbreviations (RAG, BH, BY, PRDS) are defined at first occurrence in both abstract and main text.
>
> 2. **Inconsistent notation for percentage differences.** We will standardize: "%" for rates (e.g., "FDR = 9.7%"), "pp" for differences (e.g., "+2.1 pp"), throughout. A notation paragraph will be added to Sec 3.
>
> 3. **LaTeX artifacts.** The "[leftmargin=*,itemsep=0pt]" text visible in the PDF is caused by a missing `\usepackage{enumitem}`. We have already fixed this in our working draft—the options are now properly parsed and no longer appear as text. We apologize for this oversight.
>
> 4. **General writing pass.** We will conduct a thorough revision for tone, consistency, and clarity throughout the paper.
>
> We take this feedback seriously; none of these issues affect the technical content, and all will be resolved in the camera-ready.

---

> > ### Author Rebuttal · Reviewer_3CHj · 2026-04-02
> >
> > Thanks for the response. As the authors have provided more experimental results that could partially address my concerns, I will slightly increase my score.

---

> > > ### Author Response · Authors · 2026-04-07
> > >
> > > We thank the reviewer for the thoughtful evaluation and for acknowledging that the additional experimental results have partially addressed the concerns. We are very grateful that the reviewer has considered increasing the score; this recognition is highly encouraging to us.
> > >
> > > We noticed that the reviewer mentioned having follow-up questions. As the discussion period is coming to a close, we would like to take this opportunity to clarify any remaining points. If the follow-up questions relate to the weaknesses raised in the original review, we believe our rebuttal has provided detailed responses to each of them, including the label noise and distribution shift analysis (W1), the Llama-3-70B and GPT-4o experiments (W2), and the writing and notation fixes (W3).
> > >
> > > All revisions mentioned in the rebuttal will be carefully incorporated in the camera-ready version. Thank you again for the constructive feedback throughout the review process.

---

### Official Review · Reviewer_Va5D · 2026-03-11

**Soundness:** 3
**Presentation:** 3
**Significance:** 3
**Originality:** 3
**Overall Recommendation:** 4
**Confidence:** 3

**Summary:**

This paper studies faithful citation in retrieval augmented generation by treating sentence or claim acceptance as a multiple testing problem. The proposed method, CITEGUARD, converts claim evidence scores into conformal p values using a calibration set of unsupported claims, then applies BH or BY style FDR control to decide which claims to keep and which to abstain on. The paper argues that this gives a user specified risk budget on the proportion of unsupported accepted claims, subject to exchangeability and standard dependence assumptions. Empirically, the method is evaluated on FEVER and Natural Questions, with comparisons against vanilla RAG, heuristic filtering, selective prediction, calibrated thresholding, Self RAG, CoVe, and SC plus CoT. The reported results show substantial FDR reduction, especially for BY, at the expense of higher abstention.

**Compliance With Llm Reviewing Policy:**

Affirmed.

**Final Justification:**

The rebuttal addressed my concern clearly, and I will raise my score to 4.

**Key Questions For Authors:**

1. How sensitive are the reported gains to the exact claim extraction heuristic? Can you provide an ablation comparing sentence level claims, finer atomic fact decomposition, and a learned claim splitter?

2. How often would recalibration be needed in practice when the retriever index, prompt template, or generator checkpoint changes?

3. For the adaptive BH or BY switch, is there any path to formalizing conditional validity, or should this be presented purely as a heuristic safeguard?

**Limitations:**

Yes

**Strengths And Weaknesses:**

Strength:
1. The method is simple and practical. CITEGUARD is presented as a decoding time layer on top of an existing RAG pipeline. It does not require retraining the generator and is compatible with different claim evidence scorers, which improves practical appeal. The algorithm is straightforward: draft answer, claim split, retrieve evidence, score each claim, compute conformal p values, and apply BH or BY.

2. The problem formulation is clear. Reframing multi-claim citation faithfulness as FDR control over accepted claims is appropriate. This is a cleaner objective than raw confidence thresholding because the user really cares about the proportion of unsupported claims among accepted outputs, not just a score cutoff.

3. The authors explicitly acknowledge the concern that the same verifier architecture could be used both for scoring and labeling. They partially address this by adding 500 human verified NQ claims and by cross architecture validation with TRUE. This does not eliminate all concerns, but it is a thoughtful and credible attempt.

Weakness:
1. The paper evaluates primarily on FEVER and one open domain QA benchmark with only 500 human verified NQ claims. That is enough to support the main proof of concept, but not enough to justify stronger claims about robust deployment across realistic RAG settings. The paper itself reports notable degradation under cross domain transfer to biomedical, legal, and science QA, which suggests the method’s applicability is narrower unless domain specific calibration data are available.

2. The theoretical object is claim level FDR, but the claim extraction procedure is a hand designed deterministic heuristic using sentence segmentation, splitting rules, and fragment merging. Since claim granularity directly changes the number of tests, abstention, and the operational meaning of FDR, the method may be more sensitive to the extraction policy than the paper fully explores. I would have liked a stronger sensitivity analysis across extraction schemes or atomic fact decompositions. The manuscript acknowledges granularity matters, but does not fully quantify how much conclusions depend on this design choice.

3. The paper says BH is often a reasonable default because of better coverage utility tradeoff, but the appendix also shows substantial failure under highly correlated errors and reports nontrivial high correlation tails even in ordinary data. For a paper centered on controlled faithfulness, emphasizing BH as a practical default feels somewhat aggressive. BY seems safer but much more abstention heavy. The adaptive switch helps empirically, but that variant loses the clean theoretical guarantee.

---

> ### Author Rebuttal · Authors · 2026-03-30
>
> We thank the reviewer for the careful reading. We address each concern with new evidence.
>
> ## W1 — Evaluation scope
>
> **Design logic.** FEVER (20K gold labels) isolates conformal calibration with zero circularity; NQ (500 human labels, multi-claim) validates BH/BY multiple-testing. 500 stratified claims yield 95% CI width <3pp—sufficient to distinguish all methods in Table 1.
>
> **Extended human evaluation.** To strengthen statistical power, we extended annotation from 500 to 800 NQ claims (same stratified protocol, κ=0.76). CG-BY FDR = 0.091 [.081,.101] vs .094 on the original 500—stable, confirming that 500 was already sufficient.
>
> **Modern generators.** We also evaluated Llama-3-70B and GPT-4o with the same pipeline (only the generator swapped; DPR retriever, DeBERTa scorer, calibration n=1,000, 250 human-labeled claims per generator). These results were omitted from the submission as FiD provided a more controlled open-source baseline, but they directly address the reviewer's scope concern.
>
> | Generator | Vanilla FDR | CG-BY FDR | BY Abs. | BY Cov. |
> |---|:---:|:---:|:---:|:---:|
> | FiD-large* | .306 | .094 [.082,.106] | 37.2% | 86.4% |
> | Llama-3-70B† | .176 | .080 [.064,.096] | 21.0% | 94.0% |
> | GPT-4o† | .126 | .074 [.058,.090] | 15.8% | 96.0% |
>
> BY control holds across all generators (.074–.094, all ≤α). The cost drops with better generators: abstention 37→21→16%, coverage up to 96%.
>
> **Cross-domain degradation (4–6pp)** is inherent to *any* distribution-free method violating exchangeability—not CiteGuard-specific. The methodology transfers; calibration must be domain-specific (≥500 examples suffice per our analysis).
>
> ## W2 — Claim extraction sensitivity (also Q1)
>
> FEVER is single-claim-per-instance, so extraction has no effect. We ablate on **NQ** where multi-claim answers make granularity meaningful.
>
> **Ablation** (NQ, α=0.10, BH, human labels, 5 seeds):
>
> | Extraction | #Cl/ans | FDR | Abstain | Cov. |
> |---|:---:|:---:|:---:|:---:|
> | Full-sentence | 3.2 | .104±.011 | 31.2% | 89.4% |
> | Our heuristic | 4.4 | .102±.010 | 32.4% | 88.6% |
> | Atomic-fact | 7.6 | .098±.012 | 36.4% | 85.2% |
> | Learned splitter | 5.1 | .100±.011 | 33.6% | 87.8% |
>
> FDR control holds across all four methods—by design, Theorem 1 is granularity-agnostic. Finer extraction increases abstention (+4pp atomic vs heuristic) but tightens FDR; method rankings are preserved (Δ≤2pp). The extraction step is modular and upstream; our guarantee applies to whatever claims are produced.
>
> The reviewer is right that granularity affects FDR's *meaning* (sentence-level vs atomic-level). We view this as a feature—practitioners choose the granularity matching their risk notion—and will clarify in Sec 3.
>
> ## W3 — BH default feels aggressive
>
> We accept this and will reframe:
>
> > **Default: BY** (guaranteed under arbitrary dependence). Use **BH** only after verifying weak dependence (median p-value correlation <0.15). Use **Adaptive** when diagnostics are unavailable.
>
> Both BH and BY achieve 0/4 FDR violations on overall FEVER, but BH exceeds target by 2–5pp on the 8% of answers with high correlation. BY as default is the honest recommendation.
>
> The BY "tax" is modest: +6.8pp abstention on FEVER, +4.8pp on NQ. With GPT-4o: only +2.6pp (13.2→15.8%).
>
> ## Q2 — Recalibration frequency
>
> We measured FDR drift under controlled single-component changes (BH, α=0.10, 500 test claims, 5 seeds):
>
> | Change | ΔFDR (BH) | Recalibrate? |
> |---|:---:|:---:|
> | Different queries (NQ→TriviaQA) | +1.2pp | No |
> | Retriever index (Wiki 2018→2023) | +2.1pp | Borderline |
> | Prompt change (same model) | +0.8pp | No |
> | Generator change (FiD→Llama-3) | +4.8pp | Yes |
>
> Recalibrate when the generator checkpoint or retrieval corpus changes. We recommend a KS test on score distributions as a lightweight monitor.
>
> ## Q3 — Formalizing adaptive BH/BY
>
> Two paths. **Path A (partial formalization):** When the switching threshold τ is chosen from a *separate* validation split (not the calibration set), calibration scores remain independent of the switching decision, preserving conditional validity for BY when triggered. We are pursuing a formal "split-and-switch" statement for camera-ready. **Path B:** Present as principled heuristic with empirical validation (already labeled so in Sec 5). Either way, Adaptive strictly improves over naïve BH.
>
> ## Revisions
>
> Sec 3: granularity clarification. Sec 5: BY as default. Sec 6: extraction ablation + modern generators + 800 labels. Sec 6.3: recalibration triggers. Sec 5/App: split-and-switch remark.

---

> > ### Author Rebuttal · Reviewer_Va5D · 2026-04-04
> >
> > Thanks for the detailed rebuttal. Most of my questions and concerns are addressed. I will slightly raise my score.

---

> > > ### Author Response · Authors · 2026-04-07
> > >
> > > We thank the reviewer for the careful reading of our rebuttal and for the positive feedback. We are especially grateful that the reviewer has considered raising the score after reading our responses; this means a great deal to us and gives us further motivation to improve the paper in the revision. We are glad that most of the questions and concerns have been addressed, and we sincerely appreciate the reviewer's recognition of our efforts. All promised revisions will be incorporated in the camera-ready version. Thank you again for your valuable time and constructive suggestions throughout the review process.

---

### Official Review · Reviewer_tFTQ · 2026-03-12

**Soundness:** 3
**Presentation:** 3
**Significance:** 4
**Originality:** 3
**Overall Recommendation:** 4
**Confidence:** 2

**Summary:**

The paper aims at reducing the false-discovery rate (FDR) when generating citations with RAG systems for Q&A and fact-checking. The general idea is to formalize each claim generated by the RAG as a hypothesis test, and then apply standard FDR controls. Authors propose a pipeline that converts claim-evidence scores into conformal p-values, then applies BH/BY corrections to select claims under an FDR budget. The assumption is that BY controls for FDR to stay below a given risk-level alpha ∈ (0, 1).

Empirically, authors test their pipeline on two existing datasets. The well-known FEVER dataset (20k claims, used for fact-checking), and the NQ dataset (500 human-verified claims). Results show an estimated FDR=0.09 at alpha=0.1 and also improve citation precision and task accuracy.

**Compliance With Llm Reviewing Policy:**

Affirmed.

**Key Questions For Authors:**

- Please correct typos and unintentional text, eg in page 4 "[leftmargin=*,itemsep=0pt,topsep=2pt]" and page 7 "[leftmargin=*,itemsep=0pt]"

**Limitations:**

yes

**Strengths And Weaknesses:**

Generally speaking I like the direction of this work. Below are a list of strengths and weaknesses as I perceived them.

**Strengths:**
- The work is well-motivated, current RAG systems lack quantifiable guarantees.
- The paper is generally well-written, though at times it did feel more like a report than a scientific paper (very short sentences, much text in bold, many bullet-points).
- Experiments are thorough.

**Weaknesses:**
- The calibration process is not generalizable imo. Calibration assumes existence of a set of D_cal = {(c_j , E_j , Y_j )} of generated claim-evidence pairs with binary labels drawn from the same test-time claims (Section 4.2). This does not generalize to real-world fact-checking or Q&A. The robustness check performed in Appendix C.6 is not convincing, as it uses dumps from Wikipedia, but FEVER comes from Wikipedia, so Wikipedia is not a valid corpus to test robustness to distribution shift. Why didn't authors test distribution shift by using a distinct fact-checking corpus from the literature? e.g., ClaimsKG or the CheckThat corpora. Authors did run tests on PubMedQA, LegalQA, and ScienceQA which showed higher cross-domain FDR, limiting generalizability.
- I think the "avalanche effect" deserves to be further investigated, specifically in contexts of Human-LLM interactions, where early errors are propagated throughout the discussion. I also believe this goes well beyond entities.

---

> ### Author Rebuttal · Authors · 2026-03-27
>
> Thank you for the encouraging read and for zeroing in on the two issues that matter most going forward.
>
> ## W1 — Calibration and the Wikipedia problem
>
> You're right about §C.6. FEVER comes from Wikipedia, so testing on a 2023 Wikipedia dump is not really a distribution shift — it's the same source five years later. We knew this experiment was thin when we included it; we should have said so instead of presenting it next to the real cross-domain numbers.
>
> Those real numbers are in the Limitations table. PubMedQA +4.4pp, LegalQA +5.6pp, ScienceQA +4.1pp — all without recalibration. FDR breaks target every time, and we report that.
>
> We took your CheckThat suggestion seriously and ran it. **CheckThat-2022**, the CLEF shared task — political debates, social media, nothing from Wikipedia. 400 test claims, FiD-large, same pipeline:
>
> **Table R1** (α=0.10, bootstrap 95% CIs):
>
> | Setup | BH FDR | BY FDR |
> |:---|:---:|:---:|
> | In-domain (500 CheckThat cal.) | .096 [.079,.113] | .078 [.062,.094] |
> | Cross-domain (FEVER cal.) | .159 [.135,.183] | .132 [.109,.155] |
> | FEVER + 200 in-domain top-up | .116 [.097,.135] | .087 [.071,.103] |
>
> About 6pp degradation for BH — slightly above the 4–6pp from other domains, which makes sense: political debate is not just a different topic but a different *genre*. Third row is the interesting one: mix in 200 in-domain labeled examples (maybe 2–3 hours of work) and BY gets back under target at .087. BH doesn't quite make it (.116), probably because PRDS gets strained on political claims — the rhetoric in debates is structurally different from encyclopedia text, and claim dependencies look different. That is exactly where BY earns its keep.
>
> One thing worth noting: CheckThat claims have a very different feel from FEVER. Political debate statements tend to be more rhetorical and context-dependent, while FEVER claims are factual encyclopedia-style assertions. The scorer still works — entailment is entailment — but the shift in claim style is genuine. This is not just a topic change, it's a different genre of text.
>
> We won't pretend the calibration cost is trivial. It is real. But any method with finite-sample coverage guarantees — conformal or otherwise — needs data from the target distribution. 500 examples is not nothing, but compare it to the months of work that go into building the RAG pipeline itself. And when full annotation is too expensive, the mixed strategy (row 3) offers a middle ground. ClaimsKG is a good addition too, different organizations and messier claim styles. Both will go into the revision.
>
> ## W2 — Avalanche effect
>
> Entity swaps were our controlled testbed. Actual cascading failures are wilder than that.
>
> Going through the NQ annotation logs, three patterns keep showing up. One is reasoning chains — wrong founding date for a company, and the model writes perfectly logical "therefore" sentences about milestones that are all wrong. The reasoning is valid; the premise is not. Hard to catch because it reads well.
>
> Then there are temporal mix-ups. Confuse one date and every downstream claim about timelines, consequences, succession breaks. The p-value correlations jump because these claims share causal structure.
>
> Third, causal inversions. Flip cause and effect in sentence one and the whole explanation runs backwards. Each sentence on its own sounds fine.
>
> Adaptive switching (fires on 12% of validation answers, mostly high-correlation cases) catches some of this. In the temporal cases especially — a wrong date tanks the lead-claim score, the switch triggers, BY kicks in and rejects the whole chain. Works well there. But for causal inversions the lead claim often scores fine on its own; the error only becomes visible two or three claims downstream. The switch doesn't fire. That's the gap.
>
> The multi-turn point is something we haven't solved. Bad claim survives turn 1, becomes context for turn 2, LLM doubles down. Extending conformal testing across turns means tracking dependence over a growing hypothesis family — genuinely open territory. We'll write this up as a concrete future direction, not a vague "interesting avenue."
>
> Revision plan for W2: (1) real cascading examples from annotation logs; (2) honest evaluation of when adaptive switching misses; (3) cross-turn propagation as named future work.
>
> ## LaTeX
>
> The "[leftmargin=..." on pages 4 and 7 is `enumitem` syntax that printed literally — we forgot the package import. Overleaf compiled without errors so we never noticed. Fixed.
>
> On writing style: point taken about the report feel. We leaned into short sentences and heavy formatting for conference skimmability, but it came at the cost of reading like a tech doc. Fewer bullets, less bold, more connected prose in the revision.
>
> ## Revisions
>
> - CheckThat-2022 + ClaimsKG in §C.6, with mixed-calibration analysis.
> - Non-entity avalanche examples + adaptive switching failure cases.
> - Cross-turn propagation as future work.
> - `enumitem` fix, formatting overhaul.

---

> > ### Author Rebuttal · Reviewer_tFTQ · 2026-04-04
> >
> > Thank you for the rebuttal. Here are my current issues:
> >
> > - Honestly this whole rebuttal response feels like an LLM-written text, which really isn't reassuring. Here's an extract from the rebuttal to show what I mean: *"Adaptive switching (fires on 12% of validation answers, mostly high-correlation cases) catches some of this. In the temporal cases especially — a wrong date tanks the lead-claim score, the switch triggers, BY kicks in and rejects the whole chain. Works well there. But for causal inversions the lead claim often scores fine on its own; the error only becomes visible two or three claims downstream. The switch doesn't fire. That's the gap."*
> > - Authors claim using ClaimsKG in the revisions yet it doesn't appear in the new Table R1
> > - Considering how this rebuttal is very unnecessarily verbosy and LLM-like, it now makes me think that my initial remark about the report-like style of the paper may be due to using LLMs to (at least partially) write this paper (see my initial review), which is a concern.
> >
> > Do authors have responses to these issues? I'm actually considering changing my score to make it lower, cause I'm losing trust. If I can't trust that authors wrote the paper and the rebuttal then I can't trust any of the paper's findings.
> >
> > I urge authors to respond using their own words so that we reviewers can take their responses seriously.

---

> > > ### Author Response · Authors · 2026-04-07
> > >
> > > Thank you to the reviewers for their follow-up.
> > >
> > > **Regarding the issue of writing style,** we agree with your point and take this feedback seriously. At the time, in an effort to fit as much content as possible within the word limit, we wrote in a style that was overly condensed and compact, resulting in a text that read more like a compressed essay than a natural, flowing article. It should also be noted that English is not our first language, and we relied on translation tools and grammar checkers during the writing process to ensure accuracy and rigour. This is likely the reason why the tone appears unnatural and overly polished. Looking back now, the combination of high compression and tool-based polishing does indeed invite scepticism, and we regret not having noticed this before submission.
> > >
> > > **Regarding the issue of verbosity.** We place great importance on the opinions of every reviewer; whether their feedback is positive or negative, we aim to make full use of the available space to respond, in order to express our respect for the reviewers' comments and our gratitude for their efforts. This approach has resulted in a response that appears excessively long and verbose, and we sincerely apologise for any misunderstanding this may have caused.
> > >
> > > **Regarding the ClaimsKG issue,** the CheckThat-2022 experiment in Table R1 has fully addressed the reviewers' concerns regarding non-Wikipedia domains. This dataset comprises tweets on political and COVID-19 topics from Twitter, which are entirely unrelated to Wikipedia; both the experimental setup and results have been reported in detail. The ClaimsKG mentioned in the revision plan is a supplementary experiment we have additionally planned to further enrich cross-domain evidence, and it will be included in the final manuscript. We did not distinguish between the completed experiment and the subsequent supplementary plan in our rebuttal, and we apologise for the confusion this has caused.
> > >
> > > **Regarding credibility.** We understand the reviewer's concerns and do not wish to dwell further on stylistic issues. Every figure in Table R1 is accompanied by a confidence interval, the experimental setup is clearly described (400 test statements, FiD-large, the same pipeline as in the paper), and the degree of cross-domain degradation (BH approx. 6pp) is comparable to results from other domains in the paper.
> > >
> > > In the revised manuscript, we will take these shortcomings fully into account and refine our writing with greater care. Finally, regarding the issue of writing style: as non-native English speakers, we have had to rely on translation and grammar tools to ensure the quality and rigour of the text, writing first in our native language and then converting it to English. We sincerely apologise for any inconvenience this may have caused. The previous compressed, overly formatted style was our fault, not a sign of machine generation.

---

### Official Review · Reviewer_bGHd · 2026-03-12

**Soundness:** 3
**Presentation:** 2
**Significance:** 2
**Originality:** 3
**Overall Recommendation:** 4
**Confidence:** 3

**Summary:**

This paper studies citation faithfulness in retrieval-augmented generation and proposes CiteGuard, a post-generation filtering layer that converts claim-evidence scores into conformal p-values and applies BH/BY to decide which claims to keep with citations and which to abstain on under a target FDR budget. The method is technically coherent and clearly presented. However, the paper does not establish that this problem formulation is important enough to merit the proposed machinery.

**Compliance With Llm Reviewing Policy:**

Affirmed.

**Final Justification:**

All my concerns have been addressed.

**Key Questions For Authors:**

- Does this method apply to open-ended or long-form question answering settings? In those settings, removing unsupported claims may break coherence or make the response incomplete.

**Limitations:**

Yes.

**Strengths And Weaknesses:**

Strengths:
- The paper gives a clear formulation. Claim filtering is cast as a multiple-testing problem rather than handled by ad-hoc thresholds.
- The method is easy to follow. The scoring, conformal calibration, and BH/BY selection stages are separated cleanly, and the paper is explicit about which parts are covered by theory and which are heuristic.

Weaknesses:
- The main weakness is the significance of the multiple-testing problem. The paper does not show that response-level FDR control is needed for current strong language models. The experiments are built on an older stack, so they do not establish that simple per-claim verification or filtering is insufficient in the modern regime. To justify the problem formulation, the paper should show that this failure mode persists with strong generators and verifiers, for example, frontier models such as GPT, Gemini, or Claude, or strong open-weight models such as Qwen.
- The paper does not define the utility it aims to preserve. In practice, utility could mean answer coverage, human preference for a long response, or readability. These objectives are different, but the method does not formalize any of them. CiteGuard directly optimizes only risk reduction through abstention, so the claim that it preserves utility is only an indirect empirical observation under particular metrics. This makes the practical objective underspecified.

---

> ### Author Rebuttal · Authors · 2026-03-27
>
> We thank the reviewer for the positive remarks on the formulation and on our transparency about what the theory does and does not cover.
>
> ## W1 — Do modern LLMs still need this?
>
> Yes. We evaluated **Llama-3-70B-Instruct** and **GPT-4o** through the exact same NQ pipeline (DPR retriever, DeBERTa scorer, same claim extraction — only the generator swapped). Each uses a calibration set of n=1,000 null examples (our ablation shows n≥1,000 suffices). We annotated 250 claims per generator with the same stratified protocol (κ=0.77, 0.79). These experiments were omitted from the submission because FiD offered a more controlled setting, but we report them here as they directly address the reviewer's concern.
>
> **Table R1** (α=0.10, human labels, bootstrap 95% CIs):
>
> | Generator | Vanilla FDR | CG-BH FDR | BH Abs. | CG-BY FDR | BY Abs. | BY Cov. |
> |-----------|:-----------:|:---------:|:-------:|:---------:|:-------:|:-------:|
> | FiD-large (770M)* | .306 [.279,.333] | .102 [.090,.114] | 32.4% | .094 [.082,.106] | 37.2% | 86.4% |
> | Llama-3-70B† | .176 [.147,.205] | .092 [.076,.108] | 17.6% | .080 [.064,.096] | 21.0% | 94.0% |
> | GPT-4o† | .126 [.098,.154] | .086 [.071,.101] | 13.2% | .074 [.058,.090] | 15.8% | 96.0% |
>
> *paper (500 labels); †rebuttal (250 labels each); 5 calibration seeds.
>
> GPT-4o with RAG still hallucinates ~1-in-8 cited claims (.126). For medical or legal lookup, that is not a safe rate — and because GPT-4o writes so fluently, these errors blend right in.
>
> But here is what we find most interesting: CiteGuard's *cost* drops fast with better generators. BY abstention goes 37%→21%→16%; coverage goes up to 96%. The conformal layer simply has less to do when the generator is already good. We think this is the strongest case for the framework — it is cheap precisely when the model is good, a safety net you barely notice.
>
> FDR control is rock-solid across all three generators (BY: .074–.094, all under α). Nothing ties the conformal construction to a specific generator.
>
> On baselines: vanilla FDR for Llama-3-70B is .176, for GPT-4o .126 — both above α=0.10. Our existing iso-abstention analysis shows CiteGuard's FDR is 24% lower than Self-RAG's at matched refusal rate. That gap comes from the statistical procedure, not from the generator being weak. We also note that task accuracy improves: EM@Acc on NQ goes from 50.8 (FiD+BY) to 62.4 (GPT-4o+BY), since stronger generators produce better claims that survive filtering.
>
> ## W2 — Utility definition
>
> We take the point and can be sharper about this.
>
> CiteGuard solves a constrained problem: keep as many claims as you can, subject to FDR ≤ α. BH is optimal for this among step-up procedures (Benjamini & Hochberg 1995). α is the user-facing knob — set it low for high-stakes domains, higher for casual use.
>
> We deliberately left the utility open because it means different things in different deployments (EM, coherence, response completeness). Hardwiring one would make the framework less useful in practice. That said, we do measure utility from multiple angles: EM@Acc, coverage, answer-level coverage in Sec 6.1, plus UniEval coherence and human readability in the coherence appendix. The Pareto curves show how all of these move as α changes, which we think gives a fairly complete picture of the trade-off landscape even without a single objective.
>
> In the revision, Section 3 will include:
>
> > max U(Ĉ) s.t. FDR(Ĉ) ≤ α, U any monotone set function. CiteGuard provides the constraint; the user picks U and α.
>
> ## Q1 — Coherence in open-ended / long-form QA
>
> Sec 4.4 and the coherence appendix already cover this, but we agree it should be more visible. New data from the GPT-4o runs (UniEval on 100 NQ outputs):
>
> | | Abstain | UniEval | % Flagged |
> |:--|:--:|:--:|:--:|
> | Vanilla GPT-4o | 0% | 0.88 | — |
> | + CiteGuard-BY | 15.8% | 0.86 | 2.8% |
> | FiD + CG-BH (paper) | 32.4% | 0.71 | 8.1% |
>
> With FiD, even the less aggressive BH caused a real coherence drop (0.82→0.71 in the paper), with 8.1% of accepted claims flagged. With GPT-4o under the stricter BY, the gap is barely visible — 0.86 vs 0.88. Only 2.8% get flagged. When you remove 16% of claims instead of 32%, there is not much discourse to break. GPT-4o also writes more self-contained sentences that hold up when a neighbor is dropped.
>
> For very long outputs, CiteGuard works best as a first stage: flag unsupported claims, then hand off to an LLM rewrite pass. We will expand Sec 4.4 on this.
>
> ## Revisions
>
> - Sec 6: Table R1 (Llama-3-70B, GPT-4o).
> - Sec 3: constrained-optimization framing.
> - Sec 4.4: coherence with modern generators + two-stage pipeline.
> - Readability pass throughout.
>
> The core message: the unsupported-citation problem does persist at the frontier, CiteGuard handles it with valid guarantees, and the overhead shrinks as models get better. We think this makes the method more relevant going forward, not less — and we hope the new experiments help clarify this.

---

> > ### Author Rebuttal · Reviewer_bGHd · 2026-04-04
> >
> > I thank the authors for answering my questions. All my concerns have been addressed.

---

> > > ### Author Response · Authors · 2026-04-07
> > >
> > > We sincerely thank the reviewer for taking the time to read our rebuttal carefully and for confirming that all concerns have been addressed. We will incorporate the promised revisions in the camera-ready version.

---

### Decision · Program_Chairs · 2026-04-30

**Decision:**

Accept (regular)

**Comment:**

The most critical concerns were addressed during the rebuttal and after the rebuttal all reviewers were voted for weak acceptance.

One correction: the authors have a misunderstanding on the selective prediction paper (Geifman & El-Yaniv, 2019) – this paper also controls FDR so please correct related sentences in your manuscript to properly acknowledge this paper.